# Cellular and Molecular Response of Macrophages THP-1 during Co-Culture with Inactive *Trichophyton rubrum* Conidia

**DOI:** 10.3390/jof6040363

**Published:** 2020-12-12

**Authors:** Gabriela Gonzalez Segura, Bruna Aline Cantelli, Kamila Peronni, Pablo Rodrigo Sanches, Tatiana Takahasi Komoto, Elen Rizzi, Rene Oliveira Beleboni, Wilson Araújo da Silva Junior, Nilce Maria Martinez-Rossi, Mozart Marins, Ana Lúcia Fachin

**Affiliations:** 1Biotechnology Unit, University of Ribeirão Preto-UNAERP, Av. Costábile Romano, 2201, Ribeirão Preto CEP 14096-900, São Paulo, Brazil; gabrielagonzalezsegura@hotmail.com (G.G.S.); brucantelli@hotmail.com (B.A.C.); tattytk@hotmail.com (T.T.K.); ersanchez@unaerp.br (E.R.); rbeleboni@unaerp.br (R.O.B.); mmarins@gmb.bio.br (M.M.); 2National Institute of Science and Technology in Stem Cell and Cell Therapy, Center for Cell-Based Therapy, Av. Bandeirantes 3900, Ribeirão Preto CEP 14049-900, São Paulo, Brazil; kcperoni@gmail.com; 3Department of Genetics, Ribeirão Preto Medical School, University of São Paulo, Av. Bandeirantes 3900, Ribeirão Preto CEP 14049-900, Brazil; psanches@usp.br (P.R.S.); nmmrossi@usp.br (N.M.M.-R.); wilsonjr@usp.br (W.A.d.S.J.); 4Medicine School, University of Ribeirão Preto-UNAERP, Av. Costábile Romano, 2201, Ribeirão Preto CEP 14096-900, São Paulo, Brazil; 5Department of Genetics, Ribeirão Preto Medical School, Center for Integrative System Biology–CISBi-NAP/USP, University of São Paulo, Av. Bandeirantes 3900, Ribeirão Preto CEP 14049-900, São Paulo, Brazil; 6Center for Medical Genomics, University Hospital of the Ribeirão Preto Medical School, University of São Paulo, Av. Bandeirantes 3900, Ribeirão Preto CEP 14049-900, São Paulo, Brazil

**Keywords:** deep dermatophytosis, cytokines, microRNA NGS

## Abstract

*Trichophyton rubrum* is causing an increasing number of invasive infections, especially in immunocompromised and diabetic patients. The fungal invasive infectious process is complex and has not yet been fully elucidated. Therefore, this study aimed to understand the cellular and molecular mechanisms during the interaction of macrophages and *T. rubrum*. For this purpose, we used a co-culture of previously germinated and heat-inactivated *T. rubrum* conidia placed in contact with human macrophages cell line THP-1 for 24 h. This interaction led to a higher level of release of interleukins IL-6, IL-2, nuclear factor kappa beta (NF-κB) and an increase in reactive oxygen species (ROS) production, demonstrating the cellular defense by macrophages against dead fungal elements. Cell viability assays showed that 70% of macrophages remained viable during co-culture. Human microRNA expression is involved in fungal infection and may modulate the immune response. Thus, the macrophage expression profile of microRNAs during co-culture revealed the modulation of 83 microRNAs, with repression of 33 microRNAs and induction of 50 microRNAs. These data were analyzed using bioinformatics analysis programs and the modulation of the expression of some microRNAs was validated by qRT-PCR. In silico analysis showed that the target genes of these microRNAs are related to the inflammatory response, oxidative stress, apoptosis, drug resistance, and cell proliferation.

## 1. Introduction

*Trichophyton rubrum* is the most prevalent species in human superficial mycoses, accounting for about 69.5% of all infections caused by dermatophytes [1]. Currently, the number of immunocompromised and immunosuppressed patients with deep infections caused by dermatophytes is increasing [2,3,4].

Studies on dermatophyte–host interactions are still limited but some pre-established tools exist, including animal models, culture media supplemented with protein substrates, and cell cultures [5]. Among the cell culture, the THP-1 monocyte/macrophage line is the most widely used to evaluate the host immune response and interaction with microorganisms [6]. Basically, the host immune response in the fight against dermatophytoses depends on factors such as the host response to fungal metabolites, anatomic site affected, virulence of each infecting species [1], and local environmental characteristics [7].

Macrophage membranes possess pattern recognition receptors (PRRs) that recognize mannans and adhesins present in the cell wall of *T. rubrum*, triggering the activation of nuclear factor kappa beta (NF-κB), production of IL-1β and IL-6, and phagocytosis. Some types of fungi are able to avoid mannan receptor recognition by the host, which results in deficient immune response activation, potentiating infection. Studies have demonstrated that *T. rubrum* is able to modulate the mannan receptors found on macrophages, causing an inadequate immune response and consequent chronic infection [8,9].

An important factor of the infectious process is the monitoring of reactive oxygen species (ROS) levels since the production of these molecules by phagocytic cells (e.g., macrophages) is essential for the host’s defense mechanism to fight infection. The ROS generated by phagocytic cells are important for microbicidal activity in which these radicals damage the microbial wall, generating instability and contributing to the phagocytic process [10].

MicroRNAs are small noncoding RNAs (20 to 30 nucleotides) that have emerged as important regulators of cellular processes. These RNAs are involved in post-translational regulation, which results in the degradation of mRNA or inhibition of translation [11]. MicroRNAs are associated with different important cellular processes such as inflammatory response, immunity, oxidative stress, apoptosis, nervous system, drug resistance, and cell proliferation [12]. In humans, microRNAs are considered important regulators of immune responses since they participate in both innate and adaptive immunity [13]. However, the role of microRNAs in deep dermatophytoses is still unclear.

Within this context, the understanding of the response of human cells to dermatophytes may reveal new therapeutic targets that could result in more effective therapies against dermatophytoses [14]. In the present work, we studied the co-culture of human THP-1 macrophages with inactivated, germinated *T. rubrum* conidia and analyzed the release of interleukins, ROS induction, cell viability by the lactate dehydrogenase (LDH) assay, and expression profile of microRNAs by the MiSeq technique. This experiment was carried out to increase our understanding of the cellular and molecular processes involved in the interaction of macrophages and *T. rubrum*.

## 2. Materials and Methods

### 2.1. Trichophyton Rubrum Strain, Media and Growth Conditions

The *T. rubrum* strain College of Biological Sciences (CBS) 118892 sequenced by the Broad Institute (Cambridge, USA) was cultured on Sabouraud dextrose agar (Oxoid, Hampshire, England) for 15 days at 28 °C. Microconidia of the *T. rubrum* isolate were prepared and solution (1 × 10^7^ microconidia/mL) was grown in 5 mL liquid Sabouraud medium for 7 h under gentle shaking, as described elsewhere by [15,16]. Next, the germ tubes and hyphae were centrifuged for 10 min at 4000× *g*. The culture was washed with sterile saline. The different growth stages of *T. rubrum* were then inactivated by incubation for 60 min at 56 °C [17]. Successful heat inactivation was confirmed by further culture of heat-treated *T. rubrum*, which did not grow new colonies. The inactivated germinated *T. rubrum* microconidia were called IGC and were used in the macrophage co-culture experiments.

### 2.2. Human THP-1-Derived Macrophages, Media and Growth Conditions

The human monocytic cell line THP-1 (American Type Culture Collection ATCC TIB202), derived from an acute monocytic leukemia cell line, was purchased from Cell Lines Service GmbH (Eppelheim, Germany). The cells were cultured in Roswell Park Memorial Institute (RPMI) medium (Sigma Aldrich, St. Louis, MO, USA) supplemented with 10% fetal bovine serum at 37 °C in a humidified atmosphere containing 5% CO_2_. Sigma antibiotic solution (100 U/mL penicillin and 100 μg/mL streptomycin) was added to the medium to prevent bacterial contamination. Using a hemocytometer, THP-1 monocytes were adjusted to 1 × 10^6^ cells/mL and differentiated into macrophages using 12.5 ng/mL phorbol 12-myristate 13-acetate (PMA) dissolved in dimethyl sulfoxide (DMSO) in RPMI medium for 24 h at 37 °C in a humidified atmosphere containing 5% CO_2_. After PMA induction, THP-1 cells changed morphology and adhered to the culture dish [18].

### 2.3. Co-Culture Assay Conditions

For the co-culture, *T. rubrum* IGC were transferred to 25-cm^2^ cell culture flasks containing macrophages previously transformed with PMA. The flasks were incubated in an oven for 24 h at 37 °C in an atmosphere of 5% CO_2_. The co-culture and controls (THP-1) were observed under an optical microscope (Leica DMI 6000B Optical-Fluorescence Microscope). The co-culture flow diagram is shown in Figure 1.

### 2.4. Cell Viability

The viability of THP-1 cells prior to *T. rubrum* inoculation and after 24 h of co-culture was determined by measuring the release of lactate dehydrogenase (LDH) (TOX7 kit from Sigma-Aldrich) in RPMI medium (Sigma Aldrich) according to the manufacturer’s instructions and described elsewhere [19]. Absorbance was read in a microplate reader (Elx 800 UV Bio-Tek Instruments, Inc., Winooski, VT, USA) at 490 nm.

### 2.5. Cytokine Quantification

To confirm the activation of the macrophage cells, the proinflammatory interleukin 6 (IL-6) was quantified in co-culture supernatants and incubated for 3, 6 and 24 h as described by Das Gupta et al. [20]. After the definition of the co-culture conditions, interleukin 2 (IL-2) (31–8000 pg/mL) and interleukin IL-1βeta (IL-1β) (6–7500 pg/mL) were quantified only in 24 h co-culture supernatants. Cytokines were quantified by enzyme-linked immunosorbent assay (ELISA) using Quantikine Colorimetric Sandwich ELISA assays (Peprotech^®^) according to manufacturer’s instructions.

### 2.6. Determination of Reactive Oxygen Species

Dihydroethidium (DHE, Sigma Aldrich), an oxidative fluorescent dye, was used to evaluate in situ ROS production in a six-well cell culture plate containing the co-culture as previously described [21], with some modifications. Briefly, the unfixed co-culture was washed with phosphate buffer for 5 min, incubated with 500 µL DHE (10 µmol/L) for 1 h at room temperature in a dark chamber, and washed again three times with phosphate buffer. Each co-culture was then examined by fluorescence microscopy (Leica DMI 6000B Fluorescence Microscope) and the images were captured at 400× magnification. Red fluorescence superoxide (O_2_.^-^) and non-superoxide production were evaluated using the Image J software.

### 2.7. RNA Isolation and Integrity Analysis

Total RNA was extracted using the miRNeasy^®^ Mini kit (Qiagen, Germany) according to the manufacturer’s instructions. After extraction, the absence of proteins and phenol in the RNA was confirmed in a MidSci Nanophotometer (Midwest Scientific, St. Louis, MO, USA) and RNA integrity was assessed by microfluidic electrophoresis in an Agilent 2100 Bioanalyzer (Agilent Technologies, Santa Clara, CA, USA). Only RNA with an RNA integrity number (RIN) > 7.0 was used. These RNAs were quantified in a Quantus™ Fluorometer (Promega Corporation, Madison, WI, USA) to verify if they had the adequate concentration for library construction.

### 2.8. MiSeq

MicroRNAs were obtained from the total RNA. The cDNA libraries for RNA sequencing were constructed in triplicate for each condition (cultured macrophages and co-culture) using the MiSeq v2 Reagent kit (Illumina, USA) according to the manufacturer’s instructions. The libraries were validated by quantitative PCR (qPCR) (Quantification Guide, Illumina). Single-read and paired-end sequencing were performed on the HiSeq 2000 platform (Illumina) according to the manufacturer’s instructions. The MiSeq data were deposited in the Gene Expression Omnibus (GEO) database [22] under the accession number GSE149222.

### 2.9. Sequence Data Analysis

After sequencing on the HiSeq platform (Illumina), the quality of the sequences was evaluated with FastQC software (https://www.bioinformatics.babraham.ac.uk/projects/fastqc/). Next, adaptors, low-quality regions and sequences shorter than 17 nucleotides were removed using Trim Galore! software (https://www.bioinformatics.babraham.ac.uk/projects/trim_galore/). The resulting sequences (reads) were then mapped against the hg38 version of the human genome using Bowtie software (http://bowtie-bio.sourceforge.net/index.shtml).

HTSeq software (https://htseq.readthedocs.io/en/release_0.11.1/) was used to quantify the reads mapped to regions of known microRNAs. A gene transfer format (GTF) file containing pre-microRNA annotations was used (miRbase, Release 22.1) for this purpose. The differentially expressed microRNAs were determined using the Bioconductor package edgeR (https://bioconductor.org/packages/release/bioc/html/edgeR.html) based on the false discovery rate (FDR < 0.05). The target genes of the microRNAs identified as differentially expressed were obtained from the miRTarBase (http://mirtarbase.mbc.nctu.edu.tw/php/index.php). The miRBase (http://www.mirbase.org), miRTarBase (http://mirtarbase.mbc.nctu.edu.tw/php/index.php) and PantherDB (http://www.pantherdb.org/) databases were used to discuss the relationship of microRNAs modulated in macrophages with the pathogenicity of *T. rubrum*.

### 2.10. qRT-PCR Validation

The following microRNAs were chosen for validation: miR-1291, miR-23c, and miR-6747. These microRNAs were selected because their target genes are related to pathogenicity and pathogen recognition. First, total RNA was converted to cDNA using the TaqMan^®^ MicroRNA Reverse Transcription kit and qRT-PCR was performed using Taqman^®^ MicroRNA assays, according to manufacturer’s recommendations. Gene expression levels were calculated by comparative analysis. RNU44 was used as normalizer gene as described by Gee et al. [23]. The primers and probes used were specific for miR-1291, miR-23c and miR-6747 and are available at the Thermo Fisher Scientific site (https://www.thermofisher.com/br/en/home.html).

### 2.11. In Silico Analysis

NCBI’s PubMed database was used for in silico analysis (https://www.ncbi.nlm.nih.gov/pubmed/). To discuss the relationship of these microRNAs with the pathogenicity of fungi of clinical interest, in silico analysis programs were used for the determination of microRNA pathways (http://diana.imis.athena-innovation.gr/DianaTools/index.php). For microRNAs with the highest fold change, the target genes (http://mirtarbase.mbc.nctu.edu.tw/php/index.php) and their respective biological functions (http://www.pantherdb.org/) were analyzed.

## 3. Results

### 3.1. Quantification of Interleukins in Co-Cultures of Trichophyton Rubrum with Human THP-1 Macrophages

First, to determine the most adequate time of incubation, the co-cultures were incubated for 3, 6 and 24 h. Next, IL-6, which is related to immune response activation, was quantified and we observed a higher release of IL-6 after 24 h (Figure 2A). Hence, other interleukins were quantified only after 24 h of incubation. The release of IL-1β (Figure 2B) and IL-2 (Figure 2C) was higher in the co-culture compared to control.

### 3.2. Cell Viability 

LDH quantification was used to evaluate the viability of human THP-1 macrophages in response to *T. rubrum* IGC. After 24 h of co-culture, about 70% of THP-1 cells were viable (7 × 10^5^ cells/mL). Figure 3 illustrates the results obtained for THP-1 culture (A) and co-culture of previously germinated and heat-inactivated *T. rubrum* microconidia on THP-1 (B).

### 3.3. Evaluation of Reactive Oxygen Species Production

A difference in fluorescence intensity was observed between the co-culture and control (31% versus 14%) (Figure 4A–C). This result shows that *T. rubrum* microconidia triggered a significant increase in ROS production, indicating a cellular defense response of the THP-1 cell line.

### 3.4. Differentially Expressed MicroRNAs 

Analysis of the data showed the modulation of 83 microRNAs, including the repression of 33 microRNAs and induction of 50 microRNAs (Table 1).

### 3.5. Validation of MicroRNAs by RT-PCR

We selected three microRNAs (miR-1291, miR-23c and miR-6747) for validation by RT-PCR using Taqman^®^ probes, as illustrated in Figure 5.

### 3.6. In Silico Analysis of Significantly Modulated MicroRNAs 

For microRNAs showing the most significant modulation, some target genes and pathways related to the immune system were selected in the databases and are shown in Table 2 and Table 3.

## 4. Discussion

Although infections caused by dermatophytes are usually superficial, many studies have demonstrated the capacity of these keratinolytic fungi to cause serious deep dermatophytoses. Cases of deep infections increase along with the numbers of immunosuppressed and immunocompromised patients [24] and even in individuals without immunological alterations [25]. These data confirm the inefficacy of current antifungal therapies associated with the selection of strains resistant to the pathogen [26]. However, the mechanism of the host immune response to deep dermatophytoses has not been elucidated [7]. In an attempt to better understand this mechanism, the present study described the release of proinflammatory interleukins, ROS induction and modulation of microRNAs in human THP-1 macrophages co-cultured with *T. rubrum* IGC for 24 h. This is the first study analyzing this interaction by next-generation sequencing of microRNAs.

According to Das Guptas et al. [20], increased interleukin production compared to control is an indicator of immune response activation. Quantification of IL-6 showed that *T. rubrum* IGC activated the macrophage response after 24 h. Other studies on mice infected with *Candida albicans* have suggested that IL-6 is fundamental for protection against candidiasis [27].

In the present study, *T. rubrum* IGC sensitized macrophages and provoked the release of IL-1β and IL-2. The literature also reports an increase in IL-1β release after the sensitization of murine bone marrow macrophages with *T. rubrum* [28]. In addition, IL-1β is involved in the control of the proliferation of *T. rubrum* microconidia inside murine macrophages [28]. On the other hand, [29] did not observe the release of IL-2 in macrophages stimulated with bacterial lipopolysaccharide (LPS), suggesting that the increase in IL-2 release is related to the interaction with 1,3-ß-glucans and found exclusively on the fungal cell wall [30]. In infections caused by *Aspergillus fumigatus*, IL-2 production particularly increased during the period of germination of the fungus [31]. These data suggest a correlation between IL-2 release and invasive fungal infections.

In our experimental infection tool, we found high levels of ROS and IL-1β production. It is known that ROS production destabilizes the microbial cell wall [32]. Reduced expression of IL-1β and inactivation of inflammatory pathways were observed after the inhibition of mitochondrial ROS production in lung cells stimulated with *Aspergillus* proteases [33]. A study using co-cultures of *T. rubrum* conidia with murine macrophages has shown the inhibition of interferon-γ reduced ROS production and the cells were no longer able to control the fungal load, suggesting a role of ROS in the control of deep infections [34]. We therefore suggest the effective participation of ROS and IL-1β in the inflammatory process against *T. rubrum*, in agreement with literature data [35].

MicroRNAs can control the activation of cells of the innate immune system, such as macrophages, natural killer cells and dendritic cells [20,36,37]. The expression of microRNAs enables a rapid response of cells to the presence of pathogens, intensifying inflammatory reactions by inhibiting the translation of certain genes [38]. Das Guptas et al. [20] showed that dendritic cells and human monocytes expressed miR-132 only when stimulated with germinated and inactivated *A. fumigatus* conidia, while no expression of this microRNA was observed when the cells were stimulated with LPS [39]. Differential modulation was observed of the expression of nine different microRNAs related to the immune response mediated by macrophages derived from murine bone marrow exposed to different developmental stages of *Candida albicans*.

We observed the induction of miR-23c in the deep infection tool used in this study, suggesting that this microRNA might be related to the cell response in infections caused by *T. rubrum*. This dermatophyte has been identified as a pathogen causing deep infections in patients with type 2 diabetes mellitus [4]. MiR-23c has been little explored in the literature but [40] has suggested the involvement of miR-23c in diabetic nephropathy. Skin infections are observed in 47.5% of patients with type 2 diabetes mellitus and *T. rubrum* was the predominant isolate in most lower limb lesions of these patients [41]. Amin et al. showed that miR-23c is involved in the regulation of foot ulcer healing in patients diagnosed with type 2 diabetes, in which this microRNA may act as a new regulator of the inhibition of angiogenesis [42].

The MiSeq data indicated the modulation of miR-1291. The in silico data suggest that this microRNA is involved in the modulation of heme oxygenase 1 (HO-1). HO-1 is a cytoprotective enzyme that plays a critical role in the defense of the body against injuries caused by oxidizing agents. Within this context, the inhibition of HO-1 by the pathogen may lead to dysregulation of the inflammatory response, impairing elimination of the microorganism by the immune system [43]. MiR-6747 regulates the MAPK pathway, which plays an important role in the inflammatory response. Ishida et al. [44] showed that ß-glucans present in the cell wall of *C. albicans* promoted the activation of pathways such as ROS, p38 MAPK and Nrf2, with the consequent induction of HO-1 in oral keratinocytes, demonstrating the role of this enzyme in the stress response to infection.

There are several pattern recognition receptors in macrophage membranes that recognize mannans and adhesins present in the cell wall of *T. rubrum,* but little is known about the factors that mediate the adhesion of dermatophytes [45]. Using the Dermatophyte Tandem Repeats Database (DTRDB) [46], Bitencourt et al., [47] identified four *T. rubrum* genes (TERG_08771, TERG 05644, TERG_05576, and TERG_08369) containing tandem repeat (TR) patterns that exhibited characteristics of cell wall proteins involved in adhesion [48]. TRs are short DNA sequences that are involved in a variety of adaptive functions, including the process of fungal infection [48]. The TERG_08771 gene codes a putative adhesin-like gene similar to an adhesin (MAD1) of *Metarhizium anisopliae* [49] and to a cell surface protein of *Aspergillus fumigatus* (cspA, Afu3g08990). It also exhibits sequence similarity with genes of other dermatophyte species, indicating that the repetitive units are conserved across species. This gene may play an important role in the early stages of infection since it was expressed in medium containing keratin, in an in vitro model of keratinocyte HACat that simulates superficial infection [47]. Moreover, this gene might be related to the response to antifungal agents, such fluconazol, amphotericin B and congo red [50]. Further experiments using the THP-1 macrophage co-cultivation tool with other fungal species should be carried out to address whether the response of macrophages to the dead fungal elements of *T. rubrum* may not be specific due to the similarity of the adhesins shared among the different fungi species.

## 5. Conclusions

The interaction of human THP-1 with *T. rubrum* IGC increased the release of interleukins and ROS, indicating stimulation of a cellular defense response. In addition, this interaction showed modulation of different microRNAs (miR-1291, miR-23c and miR-6747) that could be involved in the modulation of response to infection.

## Figures and Tables

**Figure 1 jof-06-00363-f001:**
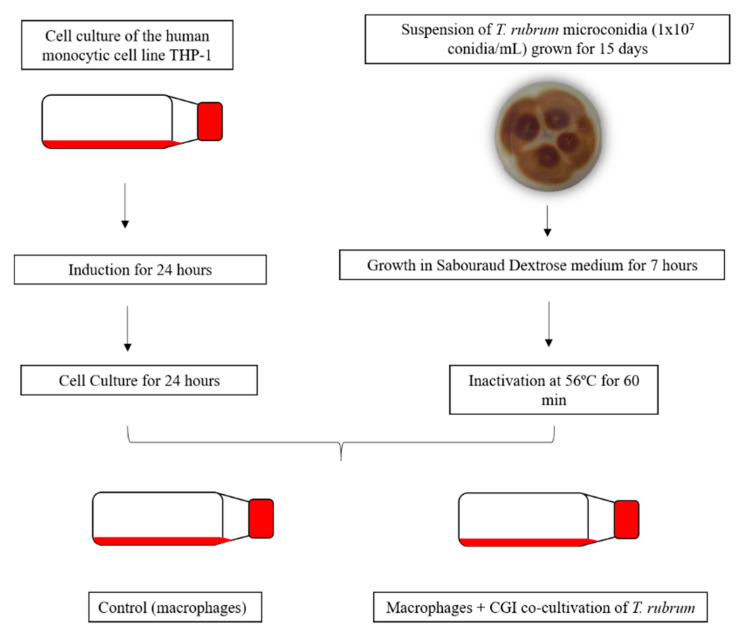
Flow diagram of macrophages and conidia germinated and heat-inactivated (CGI) of *T. rubrum* co-cultivation.

**Figure 2 jof-06-00363-f002:**
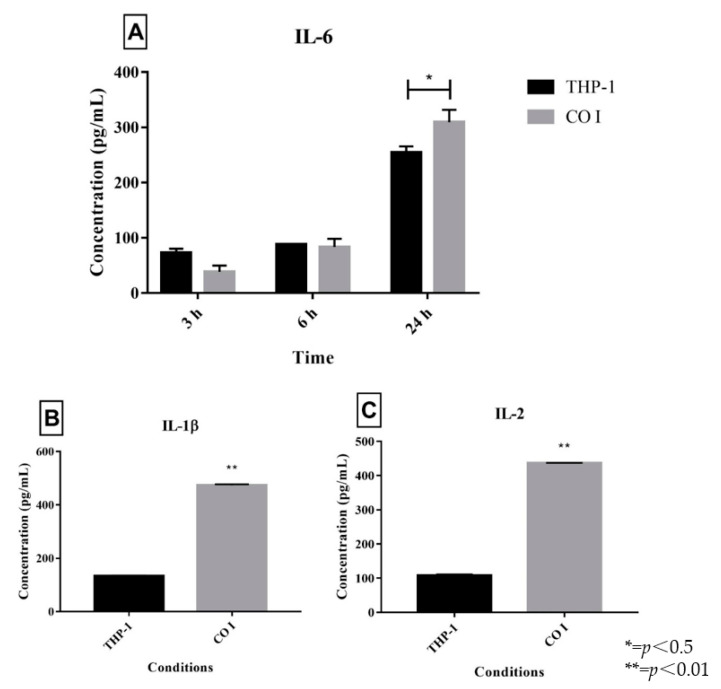
Quantification of IL-6 after different incubation times (**A**) and IL-1β (**B**) and IL-2 (**C**) in co-cultures of inactivated germinated *Trichophyton rubrum* microconidia (CGI) with human THP-1 macrophages compared to control (THP-1) after 24 h.

**Figure 3 jof-06-00363-f003:**
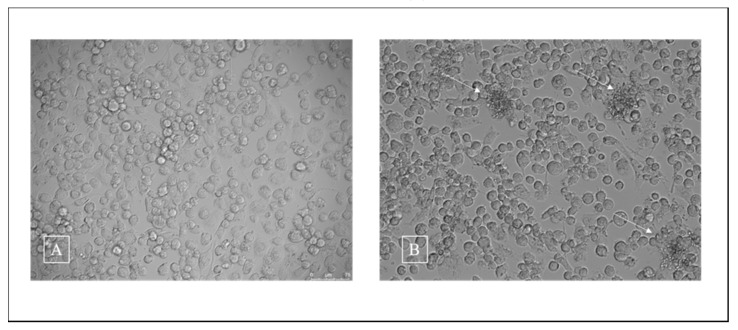
Co-culture of inactivated *Trichophyton rubrum* microconidia with THP-1 cells for 24 h. (**A**): THP-1 cell line (magnification 400×). (**B**): Co-culture. The arrows indicate fungal elements in contact with macrophages (magnification 400×).

**Figure 4 jof-06-00363-f004:**
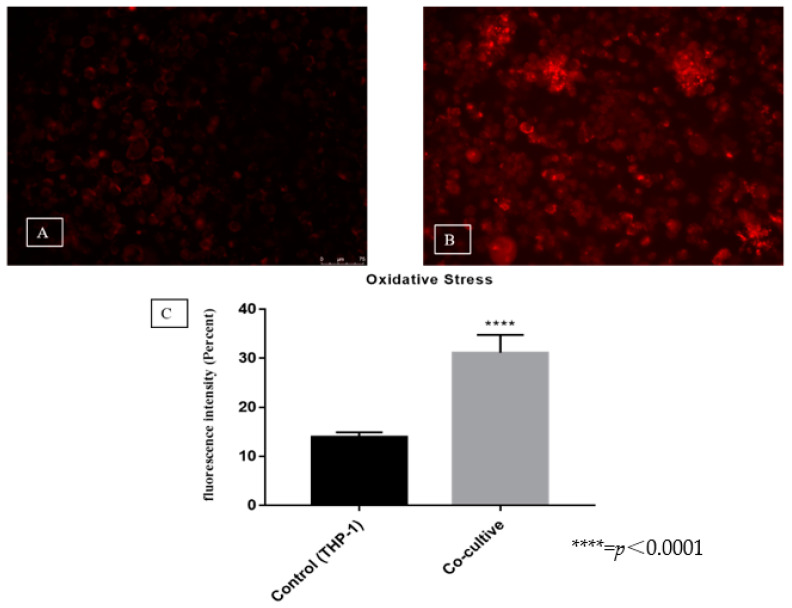
Representative photomicrograph of fluorescence intensity in the control (**A**) and co-culture (**B**). Quantification of fluorescence intensity (**C**).

**Figure 5 jof-06-00363-f005:**
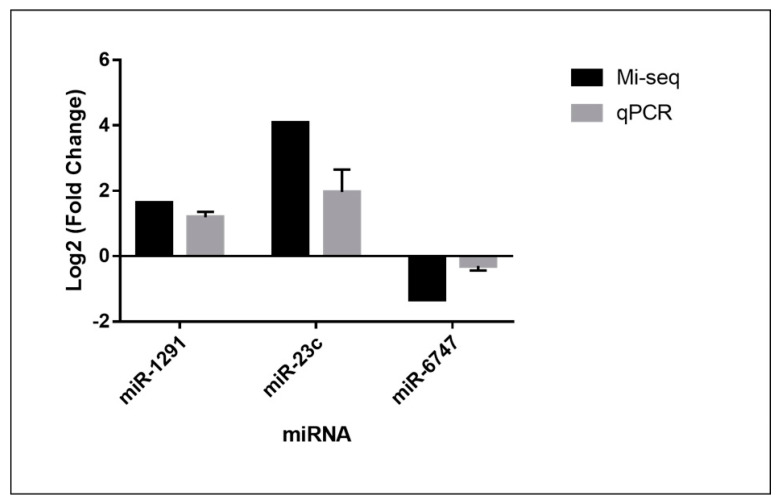
Comparison of gene modulation analyzed by microRNA-seq and quantitative PCR (qPCR). The error bars represent the standard error of three independent replicates.

**Table 1 jof-06-00363-t001:** Differentially expressed microRNAs in co-cultures of *Trichophyton rubrum* with human THP-1 macrophages.

Expression	microRNA	Log2 Fold Change	*p*-Value	Expression	microRNA	Log2 Fold Change	*p*-Value
Upregulated	hsa-miR-1244	4.58	0.012	Downregulated	hsa-let-7g-5p	−0.04	0.017
	hsa-miR-3202	4.37	0.006		hsa-let-7i-5p	−0.04	0.010
	hsa-miR-4751	4.28	0.006		hsa-miR-148a-3p	−0.06	0.000
	hsa-miR-23c	4.07	0.018		hsa-miR-450b-5p	−0.16	0.042
	hsa-miR-3680-5p	3.78	0.026		hsa-miR-19a-3p	−0.29	0.000
	hsa-miR-3129-3p	3.77	0.027		hsa-miR-10395-3p	−0.34	0.041
	hsa-miR-153-5p	3.72	0.029		hsa-miR-582-5p	−0.37	0.021
	hsa-miR-5696	3.70	0.029		hsa-miR-331-3p	−0.39	0.000
	hsa-miR-4523	3.43	0.049		hsa-miR-210-3p	−0.51	0.009
	hsa-miR-3691-3p	3.42	0.050		hsa-miR-195-5p	−0.52	0.037
	hsa-miR-4507	3.42	0.050		hsa-miR-218-5p	−0.67	0.050
	hsa-miR-1292-5p	2.57	0.019		hsa-miR-34b-3p	−0.67	0.036
	hsa-miR-16-1-3p	2.37	0.009		hsa-miR-3135b	−0.69	0.042
	hsa-miR-6810-5p	2.21	0.027		hsa-miR-1246	−0.85	0.033
	hsa-miR-1291	1.62	0.037		hsa-miR-96-5p	−0.86	0.030
	hsa-miR-378h	1.49	0.035		hsa-miR-769-3p	−0.88	0.010
	hsa-miR-4767	1.46	0.003		hsa-miR-33b-3p	−0.89	0.046
	hsa-miR-9901	1.46	0.018		hsa-miR-6793-3p	−1.05	0.021
	hsa-miR-885-5p	1.04	0.032		hsa-miR-6747-3p	−1.33	0.033
	hsa-miR-3188	0.97	0.041		hsa-miR-1908-3p	−1.37	0.006
	hsa-miR-590-5p	0.84	0.041		hsa-miR-548f-3p	−1.40	0.000
	hsa-miR-627-3p	0.82	0.016		hsa-miR-3619-3p	−1.49	0.033
	hsa-miR-133b	0.77	0.007		hsa-miR-3187-3p	−1.58	0.048
	hsa-miR-133a-3p	0.74	0.004		hsa-miR-4786-5p	−1.78	0.040
	hsa-miR-212-5p	0.69	0.000		hsa-miR-3189-3p	−1.79	0.033
	hsa-miR-1343-3p	0.65	0.028		hsa-miR-4485-3p	−2.17	0.000
	hsa-miR-224-5p	0.65	0.035		hsa-miR-10395-5p	−2.22	0.000
	hsa-miR-106a-5p	0.57	0.044		hsa-miR-548ai	−3.65	0.023
	hsa-miR-193a-3p	0.46	0.003		hsa-miR-3663-3p	−3.66	0.019
	hsa-miR-501-3p	0.42	0.009		hsa-miR-5187-3p	−3.87	0.024
	hsa-miR-6511a-3p	0.34	0.043		hsa-miR-3682-3p	−3.90	0.012
	hsa-miR-1976	0.34	0.044		hsa-miR-6876-5p	−4.11	0.005
	hsa-miR-1249-3p	0.32	0.003		hsa-miR-144-3p	−4.28	0.010
	hsa-miR-23a-5p	0.28	0.001				
	hsa-miR-29b-3p	0.27	0.001				
	hsa-miR-100-5p	0.26	0.017				
	hsa-miR-124-3p	0.26	0.026				
	hsa-miR-181c-3p	0.25	0.034				
	hsa-miR-326	0.24	0.004				
	hsa-miR-2116-3p	0.24	0.004				
	hsa-miR-1301-3p	0.22	0.007				
	hsa-miR-378c	0.17	0.001				
	hsa-miR-378d	0.16	0.000				
	hsa-miR-378a-3p	0.15	0.000				
	hsa-miR-941	0.14	0.000				
	hsa-miR-221-3p	0.11	0.001				
	hsa-miR-30e-3p	0.08	0.008				
	hsa-miR-30a-3p	0.08	0.007				

**Note:**Table 1 shows different microRNAs at *p* < 0.05.

**Table 2 jof-06-00363-t002:** In silico analysis of induced microRNAs with the highest fold change and some target genes and pathways involved.

miRNA	Target Genes	Pathways Involved
hsa-miR-1244	MAPK1-CSNK1A1-ACER2-AVPR1A1-TDGF1P3-HSP90AA1-SMAD7-ABHD2-RAB10-TMEM161B-HSBP1-UQCRB-AKR1B10	Beta cells activation; cytokine- and chemokine-mediated inflammation; integrity signaling; T cell activation; immune system; stimulus response
hsa-miR-3202	VAMP3-PLCG2-CACNB1-SDK1-UBE25-TRAF6-MYH2-ARRB2-SMARCD1-TNFSF15-FPR1-SESN2-HSPA6-CCL16-SXT7-NR1H2-PKM-MAG-JUNB-UBE4B-RNF185	5HT1, 5HT2, 5HT3, 5HT4 receptors; Beta cells activation; β1, β2 and β3 adrenergic receptors, release of corticotropin, histamine; ubiquitin; P38 MAPK; WNT; immune system; stimulus response
hsa-miR-1291	SPINT3-CHRNB2-ERN1-TAPBP-LIMD1- HO-1	Stimulus response

**Table 3 jof-06-00363-t003:** In silico analysis of repressed microRNAs with the highest fold change and some target genes and pathways involved.

miRNA	Target Genes	Pathways Involved
hsa-miR-6747	PRKX-GNB5-SYK-ITPR2-IL6R-SPIB-FOXO3-SPIC-SRF-UBE2B-TXR-HMGB1-GBP4-TLR10-SLFN13-FPR1-TLR7-BMPR1A-UGGT2-SIGLEC9-DENNDSB-GSR-PAQR7-SGTB-F2-SSR1-RBM43-FCAR-CCS-KCNMB1-	5HT1, 5HT2, 5HT3, 5HT4 receptors; β cell activation; interleukin, corticotropin release; ubiquitin; P38 MAPK; immune system; stimulus response
hsa-miR-3682-3p	MYLK-UBA6-TXLNG-SRRT-IL7R	Cytokine-mediated inflammation; ubiquitin; immune system; stimulus response
hsa-miR-144-3p	CPS1-RAC1-MAP3K4-GNG12-UBE2A-FZD6-SMARCAS-SMAD4-FBXW7-ARIDIA-ARID1B-MYCN-WNT7A-MAP3K4-PTGS2-GNG12-DAB2-ACSL4-PTGS2-DENNDSB-HSPA13-YOD1-TGFB1-IRS1-LEFTY1	Arginine biosynthesis; β cell activation; stress response; T cell activation; cytokine receptor-mediated signaling; ubiquitin; WNT; P38 MAPK; Toll receptor; MYO signaling, histamine H1 and H2; immune response; stimulus response

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
