# Peer review of "Cellular and Molecular Response of Macrophages THP-1 during Co-Culture with Inactive Trichophyton rubrum Conidia"

_jof, 2020, doi:10.3390/jof6040363_

Round 1

Reviewer 1 Report

The manuscript entitled "Cellular and molecular response of macrophages THP-1 during co-culture with inactive Trichophyton conidiaCellular and molecular response of macrophages THP-1during co-culture with inactive Trichophyton rubrum conidia" by Segura et al  demonstrated that the interaction of human THP-1 with Tricophyton rubrum IGC increased the release of proinflammatory interleukins and ROS and this indicates the stimulation of a cellular defense response. In addition, this interaction showed modulation of different microRNAs (miR-1291, miR-23c and miR-6747) that could be involved in the modulation of response to infection.

This research article is very well written and materials and methods very adequately described and clearly presented. I think that this article could be accepted for publication after some minor comments:

  1. The abstract should be shortened without loss of significance.
  2. Authors should explain in the abstract or in the introduction section was is THL-1 and ROS. It may be difficult to understand.
  3. 3. In the introduction section, 3rd paragraph authors should mention only case series and not case reports. I think it may be better to delete -references 7-9.
  4. To my opinion authors should shorten the introduction. It is enormous and I think will be difficult for the reader to focus on the main findings of the literature.
  5. References should be reduced to 30-40. More than 40 references are too many for original articles.

Author Response

REFEREE 1

Comments and Suggestions for Authors

The manuscript entitled "Cellular and molecular response of macrophages THP-1 during co-culture with inactive Trichophyton conidiaCellular and molecular response of macrophages THP-1during co-culture with inactive Trichophyton rubrum conidia" by Segura et al  demonstrated that the interaction of human THP-1 with Tricophyton rubrum IGC increased the release of proinflammatory interleukins and ROS and this indicates the stimulation of a cellular defense response. In addition, this interaction showed modulation of different microRNAs (miR-1291, miR-23c and miR-6747) that could be involved in the modulation of response to infection.

This research article is very well written and materials and methods very adequately described and clearly presented. I think that this article could be accepted for publication after some minor comments:

  1. The abstract should be shortened without loss of significance.

Author response 1: The abstract was shorted. Please see the new version of manuscript.  

  1. Authors should explain in the abstract or in the introduction section was is THL-1 and ROS. It may be difficult to understand.

Author response 2: We agree with suggestion. Please see the new version of manuscript.  

  1. . In the introduction section, 3rd paragraph authors should mention only case series and not case reports. I think it may be better to delete -references 7-9.

 Author response 3: The introduction was modified. Please see the new version of manuscript

  1. To my opinion authors should shorten the introduction. It is enormous and I think will be difficult for the reader to focus on the main findings of the literature. (Please see the lines 47-50).

Author response 4: The introduction was modified. Please see the new version of manuscript

  1. References should be reduced to 30-40. More than 40 references are too many for original articles.

 Author response 5: The reference was shorted. Please see the new version of manuscript. However, in Transcriptome results there´s the need include several manuscripts to describing the function of micro RNAs and target genes. In the other side, the referee 2 asked to discuss the specificity of fungal response to macrophages and we added more references.  

Reviewer 2 Report

Dear Authors,

in my opinion your work is very interesting in a cognitive context and contributes a lot to mycology and immunology of fungal infections. Authors report on the response of human THP-1 macrophages to T. rubrum conidia (IGC). Within this study they observed increased release of interleukins and ROS, indicating stimulation of a cellular defense response. In addition, Authors conclude that this interaction provide to modulation of different microRNAs that could be involved in the modulation of response to infection.

All the figures are appropriate for this type of article. In general, the paper has a logical flow and fit the aims and scope of the journal. The abstract well correspond with the main aspects of the work. Nevertheless, I see a few weak points in this work (given below), which I am convinced that the Authors are able to resolve.

First of all, in the substantive context, I see one weak point of this study described below.

As Authors declared they ,, ... simulated response of human THP-1 macrophages on the presence of T. rubrum cells "(after proposed correction) and analyzed the release of interleukins, ROS induction, cell viability by the LDH assay, and expression profile of microRNAs. The main question, which (in my opinion) the authors should refer to at least in the Discussion (or have additional results that can be presented in this paper) is:

1) What would human THP-1 macrophages respond to inactivated, germinated conidia of any other filamentous fungus also non-pathogenic? Is it possible that the THP-1 macrophages response on T. rubrum conidia (IGC) described in this manuscript is typical response to recognized mannans and adhesins present in the cell wall of cells any other fungi (for example even yeast cells which cell wall is rich in mannans and adhesins).

2) In the context of this question, do the Authors see a need for additional control in this manuscript, something like positive control where conidia or other fungal cells would be used? Herein, I think about cells of fungal species well known in the literature to be strongly inducing responses of human THP-1 macrophages (maybe Candida albicans cells).

As a reviewer I am also obligated to pay attention to the other less important weak points of this work and all mentioned below comments should be carefully considered.

Line 3

No spaces between ,,THP-1during”

Line 35-36

There is something missing and for me this sentence sounds better in this form: ,,After this incubation period, the level (concentration) of released interleukins IL-6, IL-1β and IL-2 and ROS produced was measured.”

Line 36-38

I cannot agree with the statement presented in the sentence (quote): ,,A higher release of these interleukins and an increase in ROS production by macrophages was observed during co-culture, demonstrating the cellular defense against fungal infection.” As you declare T. rubrum germinated conidia were heat inactivated, hence non-viable. For me, a substantive error can be avoided if you write "…cellular response to dead fungal elements."

Line 39, 41 and 42

There is no consistency in writing ,,micro RNAs" and ,,microRNAs"

Line 50

There is substantive mistake and according to many well known mycology textbooks this sentence should be replaced with (quote) ,,Fungal infections are divided into cutaneous, subcutaneous and systemic mycoses”

Line 56

There should be ,,diseases”

Line 57

There should be ,,keratinized tissues”

Line 75

,, … recognition of T. rubrum cells by the host immune cells” sounds more accurate.

Line 77

,, … pathogen-associated molecular patterns (PAMPs) on the surface of the cells of the infecting microorganism” sounds more precisely.

Line 97

,, … human cells…” sounds better

Line 99-100

The term "fungal infection" does not agree with what has actually been studied in this experiment. Rather, I would suggest revising this sentence to ,,We simulated response of human THP-1 macrophages on the presence of T. rubrum cells" or a similar sense.”

Line 103

,, … interaction of macrophages and T.rubrum cells.” sounds better.

Line 103

There are no spaces between "T." and "rubrum". Please check the entire document against the formal requirements of the journal (Instructions for Authors).

Line 107

Potato Dextrose Agar (PDA) stimulates the formation of both macro- and microconidia much better in T. rubrum. Why the Authors decided to use Sabouraud Dextrose Agar (?), on which medium conidiation is much less efficient.

Line 18-109

The Authors should specify which type, macro- or microconidia of T. rubrum were used in this experiment.

Line 110

As described elsewhere” sounds better

Line 79

,,NF-κB” - all the abbreviations used within the manuscript should be expanded on first use, even those present in the abstract. Please check the entire manuscript in this respect.

Line 120 and 121

Please standardize the units for expressing antibiotic concentrations (penicillin and streptomycin)

Figure 1

It would be good if the authors specified which kind of conidia (micro- or macroconidia, or both types together) were used in co-culture.

,,Macrophages + CGI of T. rubrum co-cultivation” sounds better

Line 140

Instead of "described in" I propose "described elsewhere"

Line 144, 268 and 292

,, … co-culture supernatants incubated for” sounds better. ,,(2014)” it is not necessary

Line 212 -213 and Figure 2

According to the Authors' declaration (line 212-213), Figure 2 contains 3 components named as Figure A, B and C. From the point of view of potential readers, it would be better to introduce additional markings within figure 2, for example A) B) and C).

Please provide spaces in the notation * = p <0.5

Line 226

I would suggest replacing the term "fungal fragments" with "fungal structures" or "fungal elements".

Line 232 Figure 4

There is ,,co-cultive” as I assume it should be ,,co-culture”. No caption for C component within Figure 4.

Line 248

,,In silico analysis of microRNAs significantly modulated” sounds better

Line 259

I would suggest a slight adjustment namely ,, … the capacity of these keratinolytic fungi to cause serious deep dermatophytoses.”

Line 304

I think, instead of ,,[60]” should be ,,Amin et al.” and citation [60] at the end of the sentence.

Author Response

Referee 2

Comments and Suggestions for Authors

Dear Authors,

in my opinion your work is very interesting in a cognitive context and contributes a lot to mycology and immunology of fungal infections. Authors report on the response of human THP-1 macrophages to T. rubrum conidia (IGC). Within this study they observed increased release of interleukins and ROS, indicating stimulation of a cellular defense response. In addition, Authors conclude that this interaction provide to modulation of different microRNAs that could be involved in the modulation of response to infection.

All the figures are appropriate for this type of article. In general, the paper has a logical flow and fit the aims and scope of the journal. The abstract well correspond with the main aspects of the work. Nevertheless, I see a few weak points in this work (given below), which I am convinced that the Authors are able to resolve.

First of all, in the substantive context, I see one weak point of this study described below.

As Authors declared they ,, ... simulated response of human THP-1 macrophages on the presence of T. rubrum cells "(after proposed correction) and analyzed the release of interleukins, ROS induction, cell viability by the LDH assay, and expression profile of microRNAs. The main question, which (in my opinion) the authors should refer to at least in the Discussion (or have additional results that can be presented in this paper) is:

  • What would human THP-1 macrophages respond to inactivated, germinated conidia of any other filamentous fungus also non-pathogenic? Is it possible that the THP-1 macrophages response on T. rubrum conidia (IGC) described in this manuscript is typical response to recognized mannans and adhesins present in the cell wall of cells any other fungi (for example even yeast cells which cell wall is rich in mannans and adhesins).
  • Author response: The discussion was included in the manuscript . (Please see lines 295-312)
  • ) In the context of this question, do the Authors see a need for additional control in this manuscript, something like positive control where conidia or other fungal cells would be used? Herein, I think about cells of fungal species well known in the literature to be strongly inducing responses of human THP-1 macrophages (maybe Candida albicans cells).
  • Author response: the suggestion is very interesting. However, at the present moment we have no agreement to perform this experiment. But the suggestion of conducting further experiments is discussed in the manucript. (Please see lines 308-312)

As a reviewer I am also obligated to pay attention to the other less important weak points of this work and all mentioned below comments should be carefully considered.

Line 3

No spaces between ,,THP-1during”

Author response: The modification was included in the manuscript

 Line 35-36

There is something missing and for me this sentence sounds better in this form: ,,After this incubation period, the level (concentration) of released interleukins IL-6, IL-1β and IL-2 and ROS produced was measured.”

Author response: The modification was included in the manuscript. Please see the new version of abstract

Line 36-38

I cannot agree with the statement presented in the sentence (quote): ,,A higher release of these interleukins and an increase in ROS production by macrophages was observed during co-culture, demonstrating the cellular defense against fungal infection.” As you declare T. rubrum germinated conidia were heat inactivated, hence non-viable. For me, a substantive error can be avoided if you write "…cellular response to dead fungal elements."

Author response: The modification was included in the manuscript . Please see the new version of abstract

 Line 39, 41 and 42

There is no consistency in writing ,,micro RNAs" and ,,microRNAs"

Author response: The inconsistency was corrected in the manuscript

 Line 50

There is substantive mistake and according to many well known mycology textbooks this sentence should be replaced with (quote) ,,Fungal infections are divided into cutaneous, subcutaneous and systemic mycoses”

. Author response: The introduction was shorted as suggestion of referee 1. This phrase was withdrawal of manuscript

Line 56

There should be ,,diseases”

Author response: This phrase was withdrawal of manuscript.

 Line 57

There should be ,,keratinized tissues”

Author response: This phrase was withdrawal of manuscript because the referee ask to reduced the introduction.

 Line 75

,, … recognition of T. rubrum cells by the host immune cells” sounds more accurate.

Line 77

,, … pathogen-associated molecular patterns (PAMPs) on the surface of the cells of the infecting microorganism” sounds more precisely.

Author response: This phrase was withdrawal of manuscript. Please see the revised version of introduction.  

Line 97

,, … human cells…” sounds better

Author response: The modification was included in the manuscript (please see the line 77)

 Line 99-100

The term "fungal infection" does not agree with what has actually been studied in this experiment. Rather, I would suggest revising this sentence to ,,We simulated response of human THP-1 macrophages on the presence of T. rubrum cells" or a similar sense.”

Author response: The modification was included in the manuscript (Please see the 79-83 lines)

 Line 103

,, … interaction of macrophages and T.rubrum cells.” sounds better.

Author response: The modification was included in the manuscript . (Please see the 82-83 lines)

 Line 103

There are no spaces between "T." and "rubrum". Please check the entire document against the formal requirements of the journal (Instructions for Authors).

Author response: The modification was included in the manuscript

 Line 107

Potato Dextrose Agar (PDA) stimulates the formation of both macro- and microconidia much better in T. rubrum. Why the Authors decided to use Sabouraud Dextrose Agar (?), on which medium conidiation is much less efficient.

Author response: The Sabouraud Dextrose Agar has been used in several published articles our research group. I consider it suitable for T. rubrum microconidia production.

Line 18-109

The Authors should specify which type, macro- or microconidia of T. rubrum were used in this experiment.

Author response: The authors used microconidia of T. rubrum. The modification was included in the manuscript

Line 110

As described elsewhere” sounds better

Author response: The modification was included in the manuscript (please see the line 90)

 Line 79

,,NF-κB” - all the abbreviations used within the manuscript should be expanded on first use, even those present in the abstract. Please check the entire manuscript in this respect.

Author response: The modification was included in the manuscript

Line 120 and 121

Please standardize the units for expressing antibiotic concentrations (penicillin and streptomycin)

Author response:We used the ready solution of Sigma with the two antibiotics penicillin (units) and streptomycin in μg/mL.  

Figure 1

It would be good if the authors specified which kind of conidia (micro- or macroconidia, or both types together) were used in co-culture.

,,Macrophages + CGI of T. rubrum co-cultivation” sounds better

Author response: The modification was included in the manuscript

 Line 140

Instead of "described in" I propose "described elsewhere"

Author response: The modification was included in the manuscript. (Please see the line 109)

Line 144, 268 and 292

,, … co-culture supernatants incubated for” sounds better. ,,(2014)” it is not necessary

Author response: These modifications were included in the manuscript

Line 212 -213 and Figure 2

According to the Authors' declaration (line 212-213), Figure 2 contains 3 components named as Figure A, B and C. From the point of view of potential readers, it would be better to introduce additional markings within figure 2, for example A) B) and C).

Please provide spaces in the notation * = p <0.5

Author response: The modification was included in the manuscript

Line 226

I would suggest replacing the term "fungal fragments" with "fungal structures" or "fungal elements".

Author response: The modification was included in the manuscript

 Line 232 Figure 4

There is ,,co-cultive” as I assume it should be ,,co-culture”. No caption for C component within Figure 4.

Author response: The modification was included in the manuscript (Please see line 210).

 Line 248

,,In silico analysis of microRNAs significantly modulated” sounds better

Author response: The modification was included in the manuscript (Please see the line 227)

 Line 259

I would suggest a slight adjustment namely ,, … the capacity of these keratinolytic fungi to cause serious deep dermatophytoses.”

Author response: The modification was included in the manuscript (Please see the line 236).

Line 304

I think, instead of ,,[60]” should be ,,Amin et al.” and citation [60] at the end of the sentence.

Author response: The modification was included in the manuscript . (Please see the line 281).